# Unfolding WWII Heritages with Airborne and Ground-Based Laser Scanning

**Kathleen Fei-Ching Sit [1], Chun-Ho Pun [1], Wallace W. L. Lai [1,\*], Dexter Kin-Wang Chung [1] and Chi-Man Kwong [2]**

[1] Department of Land Surveying and Geo-Informatics, The Hong Kong Polytechnic University, Hong Kong; kathleen.sit@connect.polyu.hk (K.F.-C.S.); chun-ho-ch.pun@polyu.edu.hk (C.-H.P.); dexter.chung@polyu.edu.hk (D.K.-W.C.)

[2] Department of History, Baptist University of Hong Kong, Hong Kong; cmkwong@hkbu.edu.hk

\* Correspondence: wllai@polyu.edu.hk

**Abstract:** Considering how difficult it is for a pin in the ocean to be found, painstaking searches among historical documents and eyewitness accounts often end up with more unknowns and questions. We developed a three-tier geo-spatial tech-based approach to discover and unfold the lost WWII heritage features in the countryside of Hong Kong that can be applied in other contexts. It started with an analysis of historical texts, old maps, aerial photos, and military plans in the historical geographic information system (HGIS) Project 'The Battle of Hong Kong 1941: a Spatial History Project' by Hong Kong Baptist University to define regions/points of interest. Then, 3D point clouds extracted from the government's airborne LiDAR were migrated to form a digital terrain model (DTM) for geo-registration in GIS. All point clouds were geo-referenced in HK1980 Grid via accurate positioning using the global navigation satellite system—real-time kinematics (GNSS-RTK). A red relief image map (RRIM) was then used to image the tunnels, trenches, and pillboxes in great detail by calculating the topographical openness. The last tier of the tech work was field work involving ground validation of the findings from the previous two tiers and on-site imaging using terrestrial LiDAR. The ground 3D LiDAR model of the heritage feature was then built and integrated into the DTM. The three-tier tech-based approach developed in this paper is standardised and adopted to streamline the workflow of historical and archaeological studies not only in Hong Kong but also elsewhere.

**Keywords:** WWII heritage in Hong Kong; geo-spatial technologies; digital terrain model-red relief image map (DTM-RRIM); LiDAR

## 1. Introduction

Heritage conservation nurtures a sense of identity and civic pride among the population. Hong Kong is a multicultural city with a complex historical experience. It was ruled by Britain from 1841 to 1997 and disrupted by Japanese invasion and occupation during the Second World War that lasted from December 1941 to August 1945. Despite Hong Kong's rich and complex history, it was not until recent decades that heritage conservation was seen as something worthy of the attention of the public and the authorities. To conserve local history and its physical remains, technological intervention is not only helpful but necessary.

Among the physical remains of Hong Kong's complex historical experience during the modern times, one of the largest (in terms of geographical span) was also the most neglected, namely the British-built Gin Drinkers Line that was built on the Kowloon Ridge in the 1930s. During the Battle of Hong Kong in 1941, the Line was a major defensive device against the Japanese invasion. The Line was significant in that it used the Kowloon Ridge north of Kowloon Peninsula to thwart any southerly assault from the New Territories. The total length of the Line was 18 kilometres, stretching from Gin Drinkers Bay at

Kwai Chung to Kam Shan (Golden Hill) in the Sha Tin District and ending at Port Shelter in the Sai Kung District. The Line consisted of four groups of pillboxes, as shown in Figure 1 [1]. Group 1, with 27 pillboxes, located in the area of Sai Kung. The 23 identified pillboxes in Group 2 were distributed along the mountain range in Kowloon. The 16 identified pillboxes in Group 3 ran from Sha Tin Tau to the Kowloon Reservoir area. Last but not least, the 27 identified pillboxes in Group 4 were distributed from Shing Mun Redoubt to Kwai Chung, where most of them were demolished as a result of urban development.

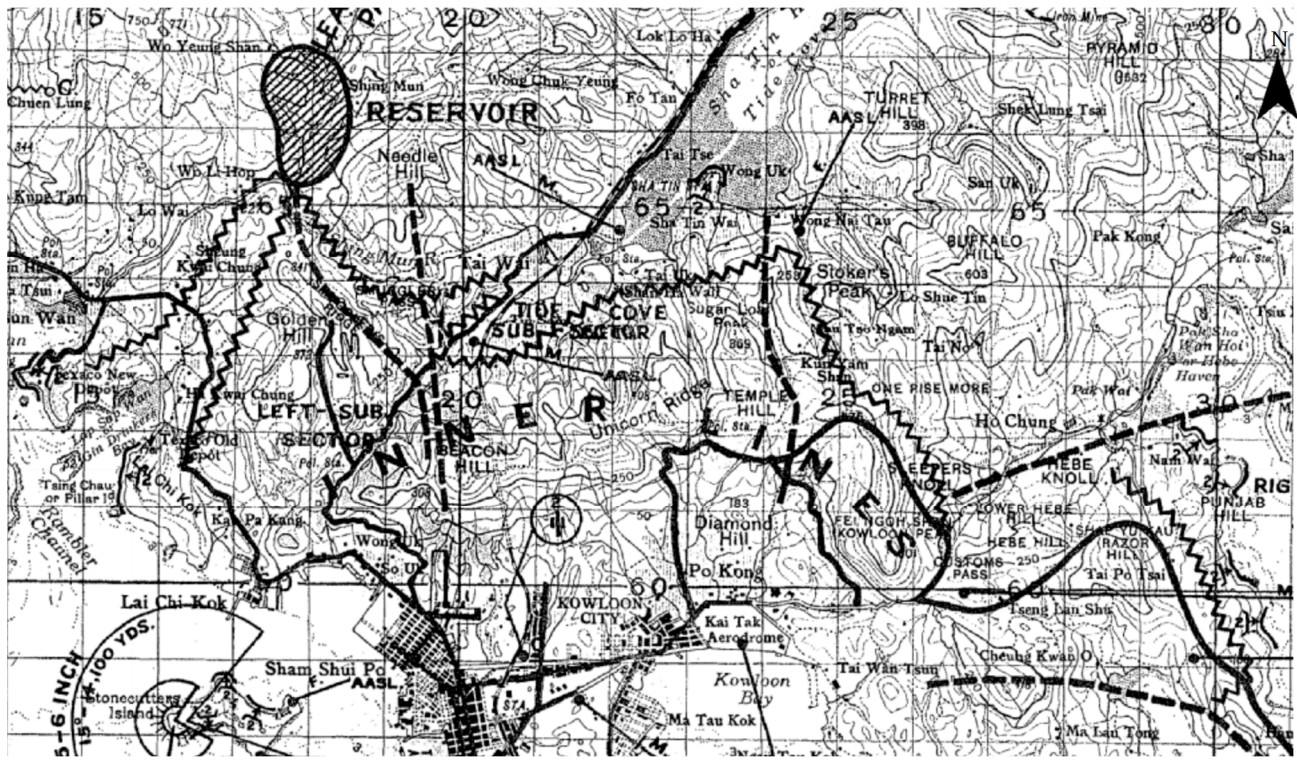

**Figure 1.** Gin Drinkers Line map [1].

Previous works have already found most, if not all, of the pillboxes of the Line [2]. However, the line consisted not only of concrete structures but also numerous field fortifications such as trenches and dug-outs that were not built with concrete, and they were often poorly documented because of their temporary nature. Moreover, after the fall of Hong Kong, the Japanese forces dug tunnels and caves near the British pillboxes in anticipation of an Allied counterattack during the final stage of the Pacific War. Because of subsequent afforestation schemes of the Hong Kong government and weathering, many of these features are now covered by trees and cannot be easily found even on the spots where they were known to be. Thus, the workflow introduced in this study helps expand and deepen our understanding of the British and Japanese military works on the hills.

In this work, we develop a three-tier geo-spatial and tech-based approach to facilitate archaeological investigation and historical interpretation of WWII in Hong Kong. Tier 1 (desktop searching the past) starts with the project 'The Battle of Hong Kong 1941: a Spatial History Project' by Hong Kong Baptist University, where historical texts, old maps, aerial photos, plans of attack and defence, and Google Earth were studied to define regions/points of interest. Tier 2 (airborne LiDAR) involves the adoption of 3D point clouds extracted from the government's airborne LiDAR ground laser return to develop a digital terrain model (DTM) for geo-registration in a geographic information system (GIS). We then explore the use of the linear and non-linear (RRIM) imaging algorithms for the best-built DTM to search for the lost and unknown WWII heritage features ('heritages') in the Gin Drinkers Line. Then together with the results from tier 1, we design a hybrid

DTM/history-based framework to prioritise and evaluate field-validation priority in the Gin Drinkers Line of defence in the Battle of Hong Kong. The output allows us to characterise and extract the DTM features of the heritages (pillboxes, trenches, tunnels, and WWII features) via field validation. Tier 3 (field Survey and validation for hybrid air–ground 3D modelling of WWII heritages) is the field work, which means a team expedition based on the setting out survey using GNSS-RTK and subsequent ground-based terrestrial/handheld LiDAR scanning after the WWII heritage features were identified in the field. All scans were tied to coded paper targets geo-referenced using GNSS-RTK. After the field work, and in the office, the built 3D point cloud and RGB models were overlaid on the DTM to generate a real 3D augmented reality of the battlefield in 1941. This paper discusses the tier 2 and tier 3 approaches in detail.

## 2. Literature Review

### 2.1. LiDAR

LiDAR makes use of laser pulses for the measurement of discrete distances and is able to produce 3D points that measure both the canopy and Earth's surface. By obtaining Point Cloud data, LiDAR can illustrate the shape of Earth by generating mesh models. A LiDAR system can scan objects by using its scanning mirror. This helps detect the particles' absorption, scattering, and emission via different wavelengths. Airborne LiDAR and terrestrial LiDAR are the two main types. For airborne LiDAR, it is composed of a flying platform, a laser scanner, a Global Navigation Satellite System (GNSS), and an Initial Measurement Unit (IMU), which are the four major elements of the airborne LiDAR system [3]. Together, these components collect the information required to produce photos and maps in high definition. An airborne platform, such as aircraft or helicopter, is required for flying a LiDAR sensor over a region of interest. This increases the scanning area and reduces time compared with terrestrial LiDAR.

After the introduction to LiDAR in the 1960s, the dependency on LiDAR to reconstruct ancient landscapes from various types of terrain has increased and has become more useful in a wide variety of disciplines [3]. The level of DTM accuracy is influenced by the point cloud data. Reconstructing archaeological sites and landscapes requires a higher point density for feature depiction, and the spacing must not be greater than 0.5 m in a square meter [4]. Through creating models, owing to LiDAR, researchers can obtain comprehensive information on individual features such as castles and coal pits that were previously hidden by overlying vegetation. Airborne LiDAR has been playing a significant role in archaeology and has brought success in investigation since the late 1990s. It was first employed in Ireland [5,6]. The survey conducted by Chase, Chase, and Weishampel [7,8] in the City of Maya, Caracol, in April 2009, that marked the debut of advanced laser technology on a large archaeological site, which was one of the earliest and most remarkable examples of LiDAR application in archaeology. Furthermore, it also mapped the terraces and presented the importance of agriculture in ancient Mayan culture, showing its ability in terrain modelling and archaeological research. In comparison with airborne LiDAR, TLS is a rather new technology for three-dimensional object modelling compared to close-range photogrammetry in the archaeological field. Ben Kacyra invented the first 3D commercial laser scanner in 1990 and founded CyArk in 2003. The well-known archaeological site of Pompeii in Italy was the first project to be recorded. Balzani et al. [9] reported in detail on the survey of the Forum of Pompeii, which was conducted using 3D Cyrax 2500 laser scanner technology, and up to 800 points/second could be surveyed with 6 mm accuracy.

Since then, the analysis of LiDAR images when researching the archaeology and geology of battlefields has raised much awareness [10]. In 2021, Adam, Storch, and Rass [11] employed an UAV-LiDAR to detect ground anomalies in Kall Trail in Germany's Hürtgen Forest. The spatial resolution (100 points/m$^2$) allowed for detection of dug-out positions and pits in a conflicted region. This enabled obtaining of higher resolution and more accurate data for validating the anomalies in this complex landscape.

Furthermore, airborne LiDAR can record radiometric information including object properties, which provides details for the archaeological features [12]. When detecting discrete objects, such as some vegetation markings caused by slightly higher levels of moisture retention or plant stress compared to the surrounding vegetation, material information contained in amplitude values obtained from a laser scan is highly helpful.

## 2.2. Visualisation Methods and GIS in Tier 2 (Airborne LiDAR)

The volume of three-dimensional topography data acquired using airborne LIDAR has greatly increased, expanding the potential of the visualisation technique. As a result of the modern LiDAR system's ability to record more data and at a higher resolution, conventional visualisation approaches such as hill-shading, contouring, and coloured relief may not be able to display comprehensive topographic data, necessitating the use of new visualisation methods [13]. Guyot, Lennon, and Hubert-Moy [14] summarised 13 visualisation techniques suitable for archaeological investigation including hill-shading [15], gradient of elevation [16], positive and negative topographic openness [17,18], local dominance [19], sky-view factor [20], RRIM [13], multi-scale topographic position [21], and the simple local relief model [22]. In this paper, we study and apply positive topographic openness, negative topographic openness, RRIM, and the sky-view factor to the various WWII heritages in Hong Kong.

### 2.2.1. Positive Topographic Openness and Negative Topographic Openness

Openness is a concept raised by Ryuzo Yokoyama, Michio Shirasawa, and Richard J. Pike [17] to represent topographic character and express the dominance or enclosure of a location on an irregular surface, which is an angular measure in terms of the relationship between surface relief and horizontal distance. Profiles along at least eight directions (N, NE, E, SE, S, SW, W, and NW) are derived from the known DTM within a radial distance for the determination of the openness value [18]. Openness is calculated by the mean of all angles from eight directions from the central point. It considers a whole sphere centred on every pixel. Positive openness (PO) can be conceptualised as the openness of the terrain to the sky. It shows features in the convex-upward direction clearly because it is the average of zenith angles subtended by a DTM point and the highest point viewed above the surface along eight directions in a particular radial distance. For example, PO helps sharpen mountain peaks and hilltops by taking the high value at the convex point. To define PO, the mean of all zenith angles from the eight compass directions is calculated using the following formula:

$$POL = (POL_0 + POL_{45} + \ldots + POL_{315})/8 \tag{1}$$

where L is the radial distance in the DTM.

Contrary to PO, negative openness (NO) is the openness below the surface. It emphasises features in the concave-upward direction such as valleys and pit, obscuring features like peaks. Similar to PO, NO is defined by the average of all nadir angles from eight directions:

$$NOL = (NOL_0 + NOL_{45} + \ldots + NOL_{315})/8 \tag{2}$$

where L is the radial distance in the DTM.

In general, convex topographic features calculate a higher score in PO value and lower NO value because the mean angle facing sunlight is greater, as shown in Figure 2. On the other hand, concave topographic features obtain a higher score in NO value and lower PO value as the mean angle facing sunlight is smaller, implying the angle below the features is greater. Eventually, PO and NO perform better in visualising peaks and valleys, respectively. It is important to stress that PO and NO are not the opposite and the inverse of each other. Figure 3 clearly shows that the NO (brown line) is not the inverse of the PO (red line), yet both of them contain useful information. As a result, the raster images of PO and NO are not opposite.

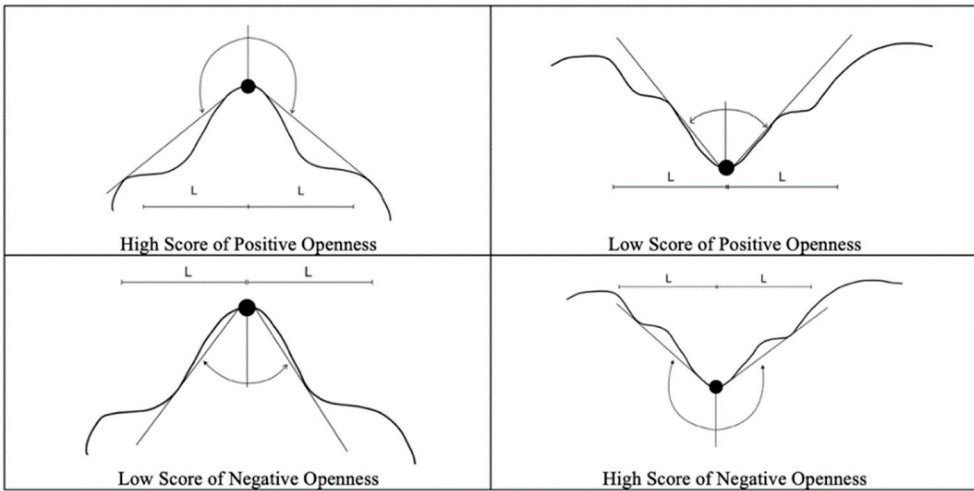

**Figure 2.** High and low values of positive openness and negative openness.

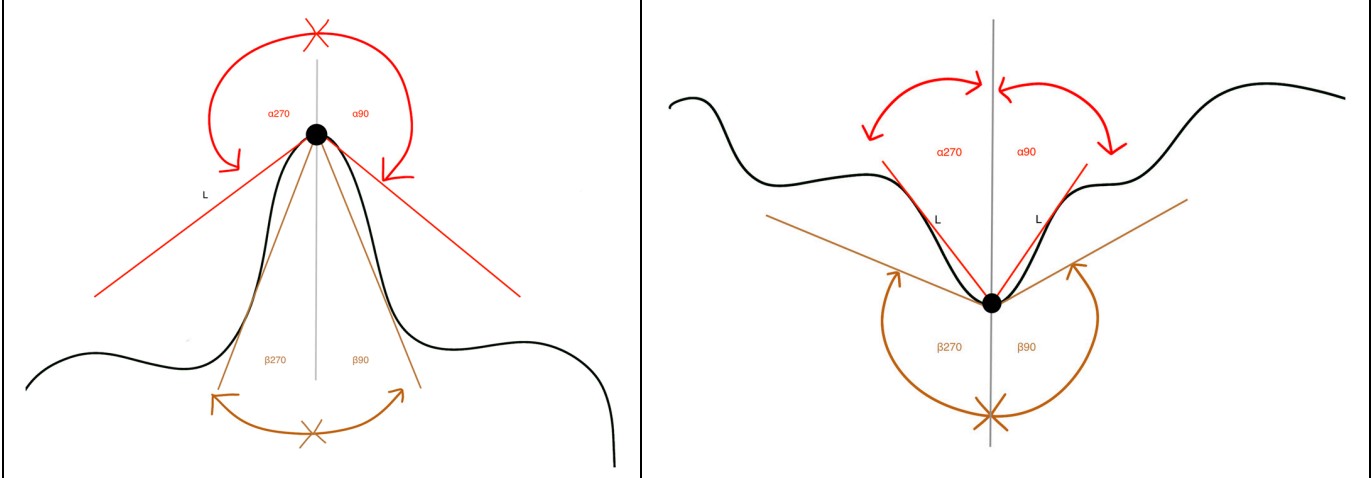

**Figure 3.** Openness calculation of positive ($\alpha$) and negative ($\beta$) along two profiles with a given length (L).

### 2.2.2. Sky-View Factor

The sky-view factor (SVF) is a ratio of the radiation emitted by the hemispheric environment to the radiation received by a flat surface. Unlike openness, SVF only uses zenith angles above the horizontal plane. The portion of the sky visible cannot be greater than a hemisphere. This is based on diffuse illumination and a parameter corresponding to the portion of visible sky limited by topography [20]. Three assumptions are made: (1) the hemisphere's whole surface is evenly lit; (2) there is no second source of directed illumination; and (3) the curvature (less than 10 km) of the Earth is neglected. This factor has advantages over other approaches such as hill-shading when used for visualisation since it displays small relief characteristics while maintaining the perception of general topography by eliminating some surface radiation and extracts new information that can be further processed [23]. Eventually, some small details in the topography can also be visualised under SVF, and it has become one of the most prominent methods in homogenising archaeologically induced microtopographic characters, which clearly distinguishes features from the surrounding terrain and delineates concave features.

The equation of the SVF can be expressed as:

$$\text{SVF} = \frac{1 - \sum_{i=1}^{n} \sin\gamma_i}{n} \tag{3}$$

where γ stands for the elevation angle of the relief horizon, and the number *n* represents the number of directions that were used to calculate the relief horizon's vertical elevation angle. SVF ranges from 0 to 1. In exposed features, values near to 1 indicate that almost the entire hemisphere is visible, while values close to 0 indicate that almost no sky is visible from the deep, bottom portions of the valleys.

### 2.2.3. Red Relief Image Map

The RRIM is initially indented to use in geomorphological interpretation using LiDAR DTM. This method is a further development of the concept of topographic openness. It is the multiple of topographic slope and openness including the ridge and valley index calculated using both PO and NO [13]. The PO takes the higher value for ridges and peaks, while the NO takes the higher value in valleys and gullies. To eliminate incident light direction dependency, the ridge and valley index, called differential openness, is a new parameter of the RRIM using the formula:

$$\text{Differiential Openness} = (\text{Positive Openness} - \text{Negative Openness})/2 \qquad (4)$$

which takes the high at the convex point and the low at the concave point, emphasising visual differences in topography.

The index is expressed using a grey-scale image layer and a red topographic slope layer on the RRIM to show the concavity and convexity of the surface without light influence. Steeper slopes are presented in brighter red while flatter surfaces are shown as grey on the RRIM. This visualisation has proved to be effective in mapping the buried remains with less than 1 m comparable height difference [24]. Therefore, the RRIM visualisation method can emphasise topographic features by showing convexities and concavities.

## 3. Methodology

### 3.1. Tier 1—Desktop Searching the Past

Only specific LiDAR data were needed because the Gin Drinkers Line is the location of interest, which indicated that pillbox information such as coordinates was crucial to exclude any extraneous data, i.e., area outside the Line. Therefore, pre-study should be undertaken for gathering useful information for further investigation. The Gin Drinkers Line was made up of 93 pillboxes, running across from Kwai Chung to Sai Kung. As the area of interest is extensive, pillbox study was necessary to comprehend the locations and specifics of the structures. The reference book Pillboxes along the Gin Drinker's Line, 80 Years after World War II, authored by Tan, Davies, and Lawrence was chosen, which recorded the coordinates (in the WGS84 coordinate system), construction details, defence area, history, and current status of all pillbox structures (Table 1). Then, the B1000 Map Index was obtained using Hong Kong Map Service 2.0 (HKMS2.0), which allowed us to view the structures in aerial photos. Photos from 1963 or 1964 had higher resolution as the flying altitude was at 3900 ft, which helped identify the existence of pillboxes, trenches, and pillbox entrances, at the time the terrains were still mostly bare ground without much vegetation.

Hidden Features Searching: In this stage, a scientific searching method was applied for investigation. First, aerial photos and old maps from the British and the Japanese were used to identify war structures and their locations. By studying the Google Earth view, the latest condition of the environment could be analysed. After obtaining the approximate location, GIS software (ArcGIS Pro 3.2) was used to analyse in order to understand the topographic features. The terrain model was presented using a chosen visualisation style for a more understandable illustration with ground classification. Vegetation removal enabled the display of the clear terrain and topographic details. This stage would focus on looking into unusual topographic features like holes or structures because they might have historical significance relating to WWII and the defined region of interest.

**Table 1.** Information regarding PB315, PB106, and PB126.

| Pillbox: | PB315 | PB106 | PB126 |
|---|---|---|---|
| Location: | The knoll to the southwest of the Kowloon Reservoir Dam | To the south of Chuk Kok Hill | Tate's Pass |
| GPS (WGS84): | 22°21′0.08″ N 114°9′8.52″ E | 22°20′40.98″ N 114°15′41.36″ E | 22°21′24.56″ N 114°13′20.03″ E |
| Construction details: | A large three-loophole PB linked by a tunnel to a concrete entrance trench | A two-loophole PB linked by a brick tunnel | A large three-loophole PB, camouflaged as a big rock, with a concrete entrance trench behind |
| Status: | The PB, its entrance trench, and tunnel are still intact. This is the only PB that remains intact in Gin Drinkers Line | PB walls and the entrance trench remaining | PB walls remaining |

Old Maps Analysis: Materials such as old British and Japanese military maps could be used to search for special features. Military maps from 1939 and 1941, provided by the Battle of Hong Kong Spatial History Project, showed the marking of military facilities. With the reference to old materials, the accuracy of finding unknown war features was higher.

Aerial Photos Analysis: Aerial photographs were geo-referenced to show the changes to the location of interest throughout 50 years. Additionally, they served as proof of the environment and status in a particular year. With the reference of aerial photos, the latest environment could be estimated for field validation.

Feature categorisation could be performed using GIS software, owing to the high resolution of the RRIM. Graves and war foxholes caused the most confusion because of their similar representations. Nonetheless, the RRIM, which was able to display the intricate characteristics and designs of tombs, made it possible to classify them. Figure 4 was a demonstration of graves and foxholes visualised using the RRIM. The shape, design, steps, and orientation of tombs were apparently shown. Eventually, graves could be classified separately from foxholes and caves. Both 2D and 3D views facilitated the investigation using GIS software. Following the marking of a pillbox's coordinates on the RRIM, the area around the pillbox would be examined for any unusual topographic characteristics within a 300 m searching radius. The feature would be targeted for closer examination if it exhibited unnatural traits.

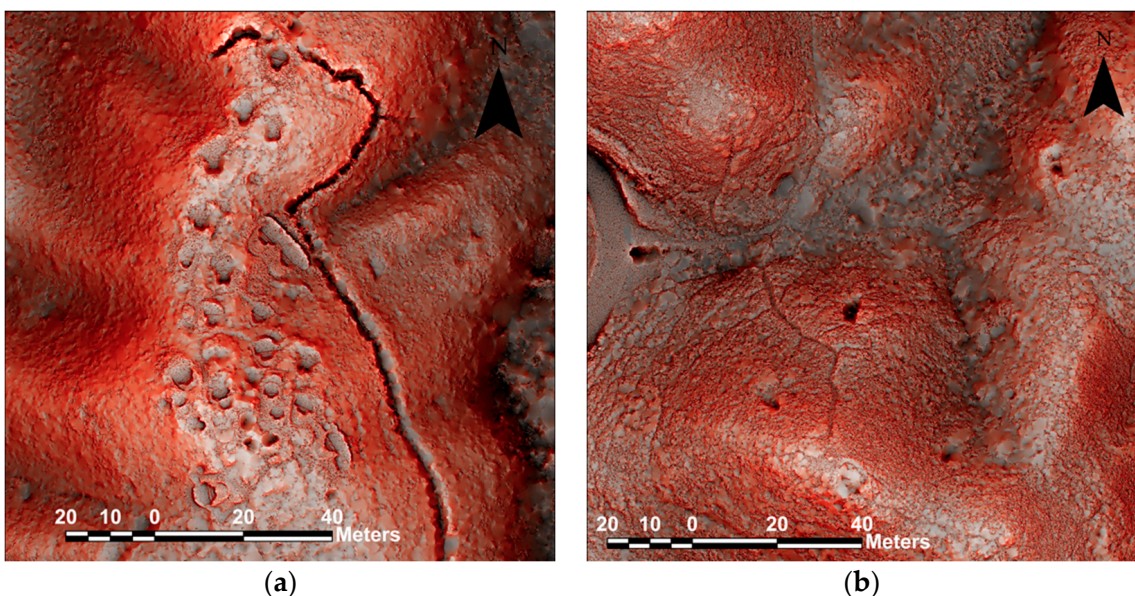

**Figure 4.** Graves and holes visualised using the RRIM (3D). (**a**) Graves. (**b**) War foxholes.

*3.2. Tier 2—Selection of Visualisation Methods*

In the tier 2 study (airborne LiDAR), there were five types of visualisation methods. Linear PO, NO, sky-view factor, mesh, and non-linear RRIM are discussed in this section, and the most suitable technique was selected from among linear and non-linear imaging of the WWII heritage features. The first four visualisation methods were applied to the ALS-derived geo-referenced raster DTM, which is a representation of the ground topographic surface of the Earth, at 5 cm resolution, with specific calculation and visualization parameters using the GIS software ArcGIS Pro and open-source tools such as QGIS, the Relief Visualization Tool, and SAGA GIS provider. The ALS-derived geo-referenced raster DTM was generated using CEDD Airborne LiDAR Point Cloud, and it filtered out all vegetation point cloud and building point cloud data, so only ground point cloud data were kept as most WWII remains were hidden in the forest. Mesh was generated via CEDD Airborne LiDAR ground point cloud using Point Cloud software Cyclone 3DR 2023.1 with an extraction grid size of 5 cm. This simplified the terrain formation, and the topographic details could be seen more obviously as there were no unnecessary data. The flying altitudes for PB315, PB106, and PB126 were approximately at 2000 ft, 2000 ft, and 3000 ft, respectively. The LiDAR data provided an average of 16 points/m², implying an average point spacing of 0.25 m. However, on occasion, this value can escalate to as high as 100 points/m². Therefore, 5 cm resolution sufficiently reflects topographic details of researched area.

The first and second methods are positive openness and negative openness. Openness is one of the linear visualisation techniques for archaeological interpretation of the digital model derived from airborne LiDAR. Positive openness obtains a high value in peaks and ridges while negative openness acquires high values in valleys. The algorithm of openness has four parameters that could be changed, which were radial limit, method, multi-scale factor, and number of sectors. The radial limit was 50 m, and the others remained the default. The radial limit controls how far each cell's openness is traced. A smaller radial limit gives a very crisp image, which is useful for very detailed mapping such as landslide scars. To secure the level of precision, the 50 m radius was found suitable for visualising the details of the terrain since only the ground point remained after filtering. The third method is sky-view factor. The range of the sky-view factor is between 0 and 1. If the value is close to 1, the entire hemisphere is visible, implying features such as peaks are exposed. However, if the value is close to 0, this implies features are deep, such as with valleys or sinks, where no sky is visible. The fourth method is mesh. A polygon mesh is made up of vertices, edges, and faces. After achieving data from the above methods,

the results can be compared. The fifth method is using the RRIM. The RRIM is the non-linear presentation, formed from differential openness overlaying the slope map with a red colour. This emphasises and sharpens the topographic changes and terrain features. The results were obtained in a darker colour layer, and the redder the colour, the steeper the slope.

A site with a WWII trench dug by the British garrison force in Sai Kung of Hong Kong was selected to compare five visualisation methods as shown in Figure 5. All cell sizes of LiDAR were in 5 cm. The Y-shaped trench could be seen in all five visualisations. The mesh method's depiction of the trench was the least distinct because the colour contrast showed the fewest variations. Negative openness was more able to reveal characteristics and depicted a clearer trench/grave than positive openness because the negative angles are greater than the positive angles due to its concave relationship to the sky. Whereas the centre part of the trail could not be seen in negative openness, roads and trails were sharpened in positive openness because of its slight projection on the ground making a more convex relationship to the sky. The sky-view factor could show graves and trenches in good condition, but it could not clearly display holes or steep slopes. The four linear methods above were outplayed by the non-linear RRIM, which overcomes all shortcomings of the linear methods because of the collected consideration of differential of openness (WWII features and graves) with the slope map overlaying (terrain) described in Section 2.2.3.

A closer look at these features is presented in Figure 6, which zooms-in on the details of each visualisation technique. In terms of trail (feature 1), negative openness was unable to visualise the centre portion of the feature on a map, making it impossible to portray a clear route. In mesh, where the link to the main road was obscured, it was similarly challenging to view the trail. Positive openness, RRIM, and sky-view factor performed better by showing an identifiable trail on map. In terms of graves (feature 2), positive and negative openness as well as mesh were unable to display the graves' pits in the earth in detail. The graves could be identified using RRIM and sky-view factor, but RRIM provided a greater level of quality by displaying the graves' sizes. Non-linear imaging (RRIM) could also provide a better visualisation for the recognition of features. In terms of slope (feature 3), where RRIM provided the most evident terrain variations, both positive and negative openness were unable to display the slope and terrain. In terms of cemeteries (feature 4), mesh's rendering of the cemetery on the left missed out on small graves' intricate characteristics. Only large graves could be depicted clearly on the map, where the visualisation of positive openness was flat regarding showing height changes. The sky-view factor and negative openness had similar effects in that the shape of the graves could be exhibited, though the former had a stronger visual effect but was still less obvious than RRIM. Non-linear imaging could exhibit the slope steepness using a vivid red colour, or other colour, and gave superior visual impact compared to grey scale when identifying changes of terrain. So, RRIM was found to be the most suitable method of producing a high-quality visual effect to strike a balance between locating WWII features and the characteristics of terrain changes. Therefore, RRIM was selected to visualise LiDAR data in GIS software in the following section. The effects of five methods were ranked as follows (Table 2): RRIM > sky-view factor > positive openness > negative openness > mesh.

**Table 2.** Summary table of methods.

| Method/Feature | Trail (1) | Grave (2) | Slope (3) | Cemetery (4) |
|:---:|:---:|:---:|:---:|:---:|
| Positive openness | High | Medium | Low | Medium |
| Negative openness | Low | Low | Low | Medium |
| Sky-view factor | High | High | High | Medium-High |
| Mesh | Low | Low | Medium | Low |
| Red relief image map | High | High | High | High |

High: clearly see details. Medium: some details are unclear. Low: half feature is blurry.

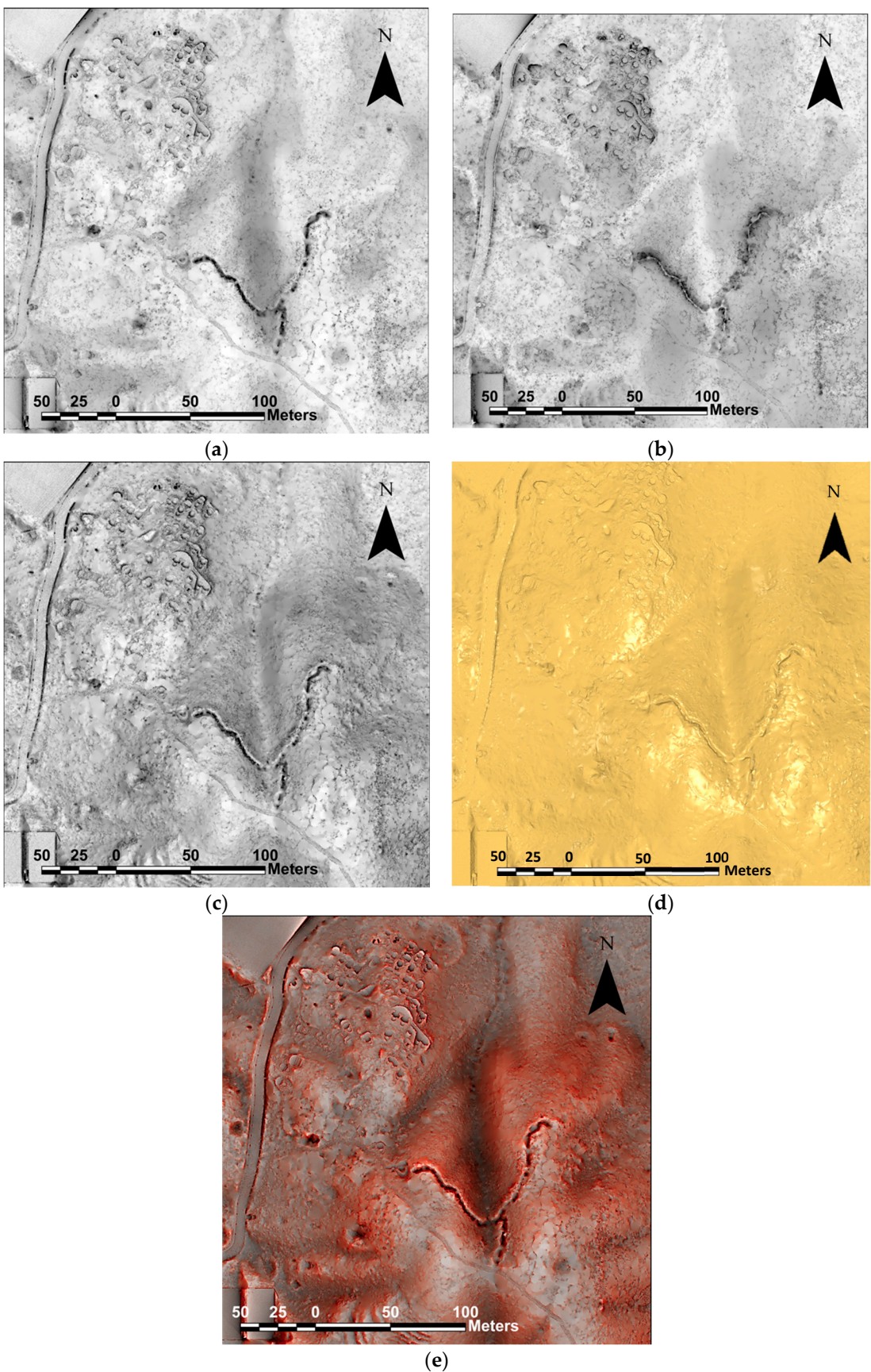

**Figure 5.** Comparison of each method. (**a**) Positive openness (linear). (**b**) Negative openness (linear). (**c**) Sky-view factor (linear). (**d**) Mesh (linear). (**e**) Red relief image map (non-linear).

Not every place is worth a visit and is safe to visit. Priority for field surveys and validation was given to the war heritages that are clearly visualized as tunnels, trenches, and pillboxes in the DTM-RRIM. The priority was classified into three classes, high, medium, and low, based on judgements from three aspects: surveyors' views on DTM-RRIM, historians' points of view, and accessibility/safety. Figure 7 presents the general results of the topography of the entire Gin Drinkers Line on the RRIM where priority of visits is given to all identified heritage features.

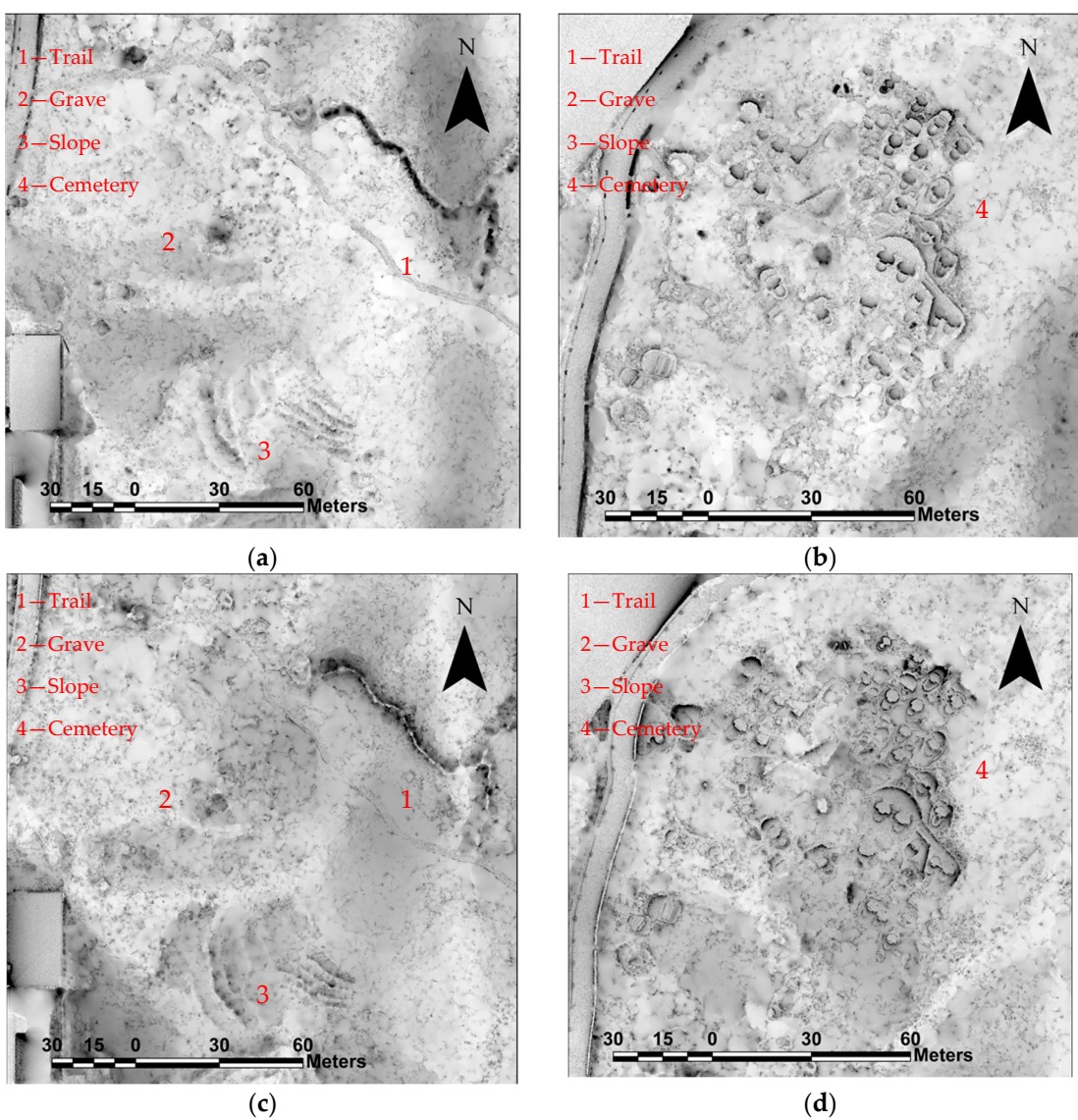

**Figure 6.** *Cont.*

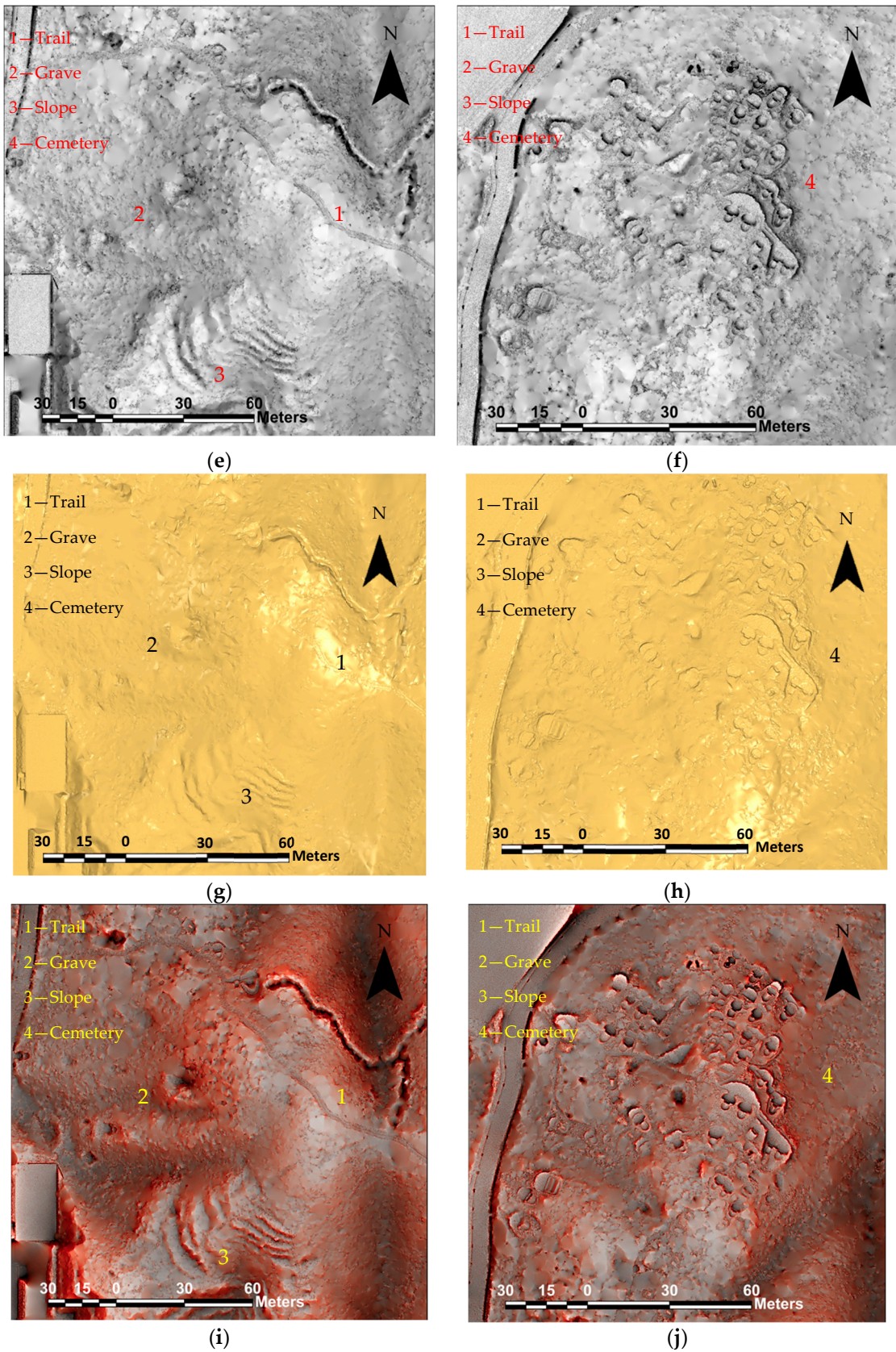

**Figure 6.** Details of each method. (**a**,**b**) Positive openness. (**c**,**d**) Negative openness. (**e**,**f**) Sky-view factor. (**g**,**h**) Mesh. (**i**,**j**) Red relief image map.

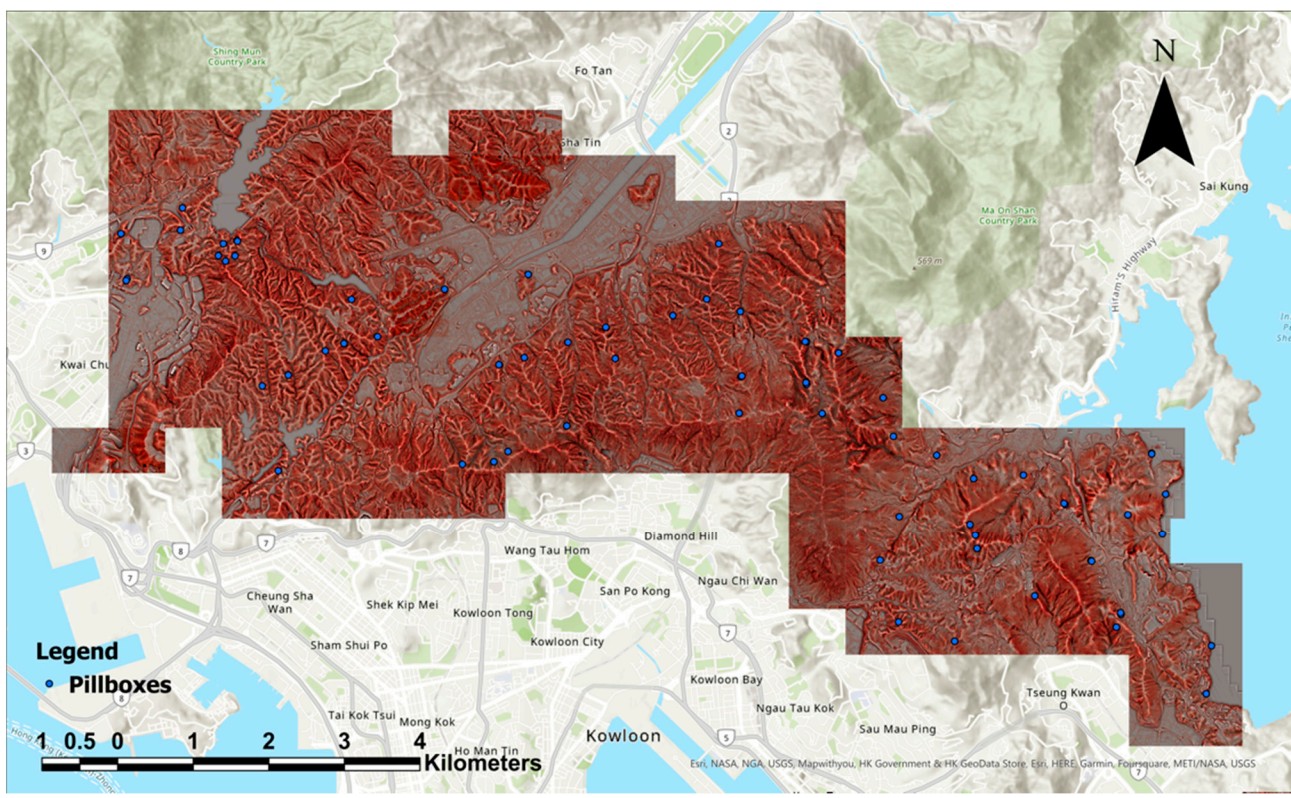

**Figure 7.** The 93 pillboxes along the Gin Drinkers Line on RRIM (2D).

### 3.3. Tier 3—Field Survey and Validation for Hybrid Air–Ground 3D Modelling of WWII Heritages

A number of geo-spatial technologies were employed in the field survey to collect data after the pillboxes and the unknown features had been identified in the DTM-RRIM. The acquisition was conducted via terrestrial laser scanning geo-referenced using GNSS-RTK for collecting accurate coordinates that could be put to maps. Two systems, namely Leica Zeno GG04 Plus and CHCNAV i90, were employed. The horizontal accuracy of GNSS-RTK within 5 cm was defined as 'fixed' in this work according to the Survey and Mapping Office of Lands Department of HKSARG. The GNSS measurement of coordinates was used to establish a few ground control points for geo-referencing the point clouds obtained using the terrestrial laser scanner (Leica RTC360) and handheld laser scanner (Leica BLK2GO). Terrestrial laser scanning was adopted for scanning features on flatter surfaces like tunnels, pillboxes, and trenches. As the terrain of the region of interest was mostly complicated and open to sky, multiple setups of Leica RTC360 were essential in order to obtain a clearer and more accurate dataset. BLK2GO was employed as a handheld scanner for operation in undulating and poor terrain, which did not favour the positioning of terrestrial laser scanner. This reduced scanning time and was more convenient in scanning rugged terrain although accuracy and precision are the trade-off. With such technologies, all the heritages were 3D scanned using the laser scanners, and the resulting point cloud were further analysed and re-constructed with the following few steps in lab: (1) cloud optimisation, (2) visual alignment, (3) manual cleaning of unwanted point cloud, and (4) geo-referencing with field GNSS-RTK coordinates. Two types of software, Leica Cyclone Register 360 2021.1.2 and Cyclone Register 360 PLUS 2023.0, were used to complete the data post-processing and visualised in Cyclone 3DR 2023.1 and ArcGIS Pro 3.2. DTM-RRIM and the terrestrial 3D point cloud models were then integrated to visualize both the terrains and the heritage features and bring the audience back to the battlefield in 1940s.

## 4. Results

Four WWII features (Figure 8) are highlighted in this paper: (I) PB315 in Kowloon Byewash Reservoir, (II) the Y-shaped trench near PB106 in Pak Shui Wun, (III) the tunnel near PB126 in Kowloon Peak, and (IV) the cave near PB126 in Kowloon Peak.

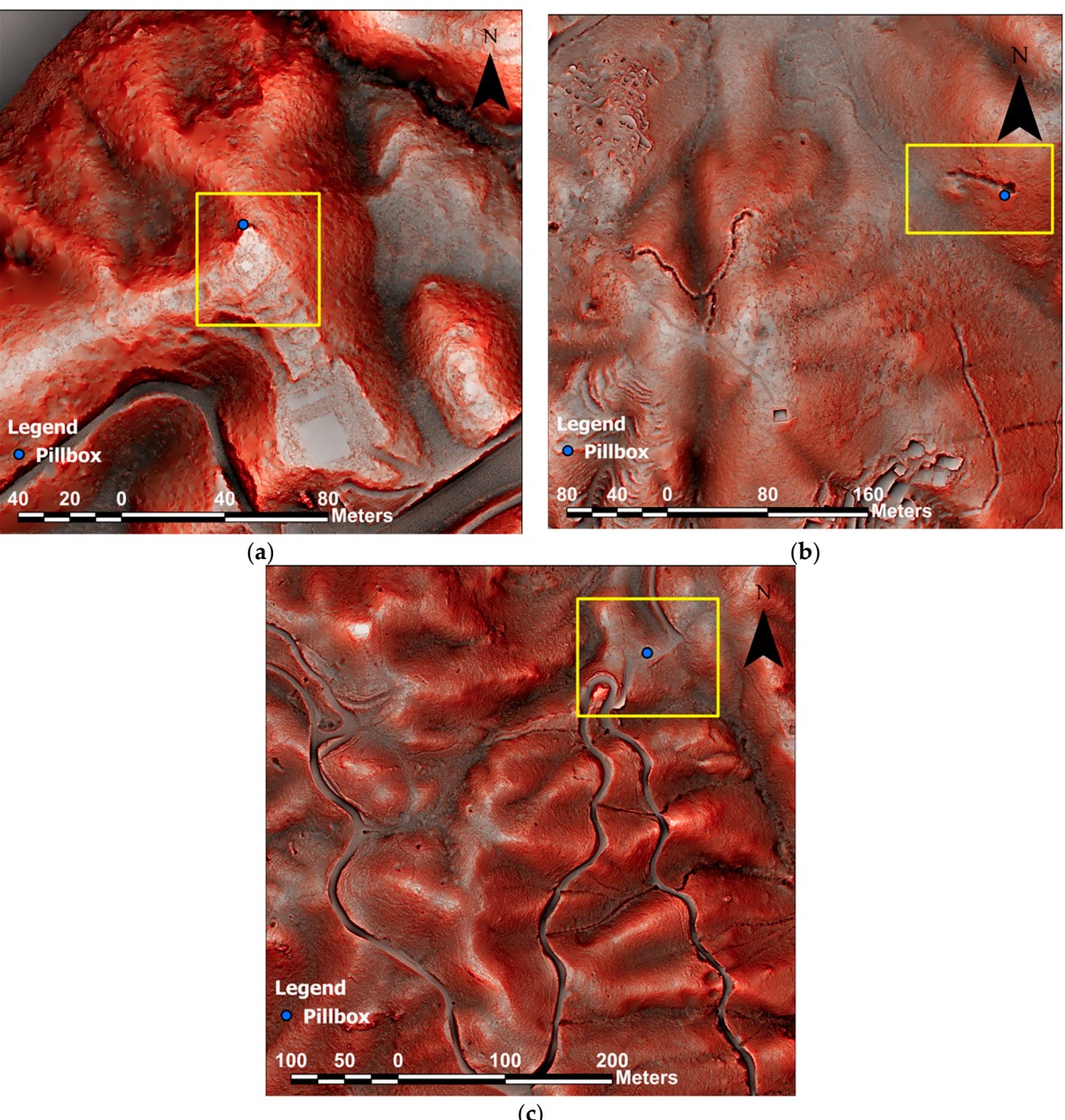

**Figure 8.** Locations of pillboxes on RRIM (2D). (**a**) PB315. (**b**) PB106. (**c**) PB126.

*4.1. WWII Feature I—Pillbox PB315 in Kowloon Byewash Reservoir (Figure 9)*

The search for hidden features in the vicinity (i.e., approximately 300 m in radius) of PB315 was conducted using tiers 1–3 analyses. According to military maps in 1939 (Figure 9a) and 1941 (Figure 9b), there was no special military feature near PB315. The aerial photo in 1964 of PB315 (Figure 9c) was investigated to understand the site condition, which indicates that the area was not completely covered by vegetation in 1964. Google Earth (Figure 9d) and RRIM (Figure 9e) do not reveal that PB315 is the only distinct feature and the only point of interest. The field visit and survey reveal that PB315 was large and well-maintained in general (Figure 10a), and two-thirds of it were buried in

the ground (Figure 10b). Therefore, PB315′s DTM-RRIM results are not as significant as those of the other WWII features II to IV. It consisted of three firing platforms, three chambers (front, main, and rear chambers), a tunnel, and an entrance trench. GNSS-RTK and RTC360 assisted in scanning the pillbox. Three checkerboard targets were positioned near the entrance for geo-referencing and positioning the terrestrial LiDAR scanning results in 14 scanning setups in total (Figure 9f).

*4.2. WWII Feature II—The Y-Shaped Trench near PB106 in Pak Shui Wun (Figure 11)*

The yellow box in Figure 11a marks the Y-shaped trench located in the southwest part of PB106 on the 1939 Japanese military map. The red dot marked by the Japanese 'ト ー チ カ' in the legend means 'pillbox'. It was possibly marked by the Japanese military for a British pillbox they found. The aerial photo in 1963 (Figure 11b) clearly showed the location of the pillbox and the unidentified Y-shaped trench, which was covered by overgrown vegetation in 2016 (Figure 11d). It was exceedingly doubtful whether the Y-shaped feature is related to PB106, which is about 250m away (Figure 11e,f). Next, the XY coordinates of the feature of interest were plotted on the DTM-RRIM in ArcGIS (Figure 11g) as a clear reference for the fieldwork investigation in tier 3. The expedition team found the Y-shaped trench at the exact coordinates indicated in Figure 11g. A terrestrial LiDAR survey with 23 scanning setups was conducted, connected by 22 paper targets for enhancing accuracy of manual alignment and providing geo-referencing in the Y-trench (Figure 11h). The numbers of targets and setups are much more than those of feature I because of its rugged terrain and overgrown vegetation (Figure 12a,b), which obstructs line of sight easily. When we look back at Google Earth (Figure 11c), it is apparent that the entire area was obscured by foliage, so that the battle structures were little known to the public.

*4.3. WWII Features III and IV—Tunnel and Cave near PB126 in Kowloon Peak (Figure 13)*

PB126 is another site close to Kowloon Peak. There were no distinctive military features close to PB126 according to military maps in 1939 (Figure 13a) and 1941 (Figure 13b). The aerial photo in 1963 (Figure 13c) clearly showed the location of the pillbox and some holes. According to Tan et al. [2] only some pillbox walls were remaining. The entrance trench was destroyed or collapsed after WWII. Yet, the pillbox was still visible in the 2021 aerial photo (Figure 13d) since it was close to the road and exposed to the sky. The holes on the left could be seen clearly in the 1963 aerial photo (Figure 13c), but they were covered by vegetation in 2021 (Figure 13d). There were some holes visible on the RRIM within a 300 m searching radius (Figure 13f,g), which could be the entrance of underground war tunnels. While the hiking paths on the hill were clearly visible in the 2017 satellite view (Figure 13e), the holes near the paths were not obvious.

Then a field-validation visit was conducted using GNSS-RTK. We found a tunnel and a cave in exactly the locations suggested in the DTM-RRIM, as shown in Figure 14a–c. The tunnel was about 1 m wide and around 1.2 m high. Because of the tunnel's width, a stationary TLS is not favourable in the confined space of tunnel, so a portable TLS scanner was used to capture point cloud data with the help of torches for collecting not only point cloud but also RGB colours for subsequent modelling (Figure 13h).

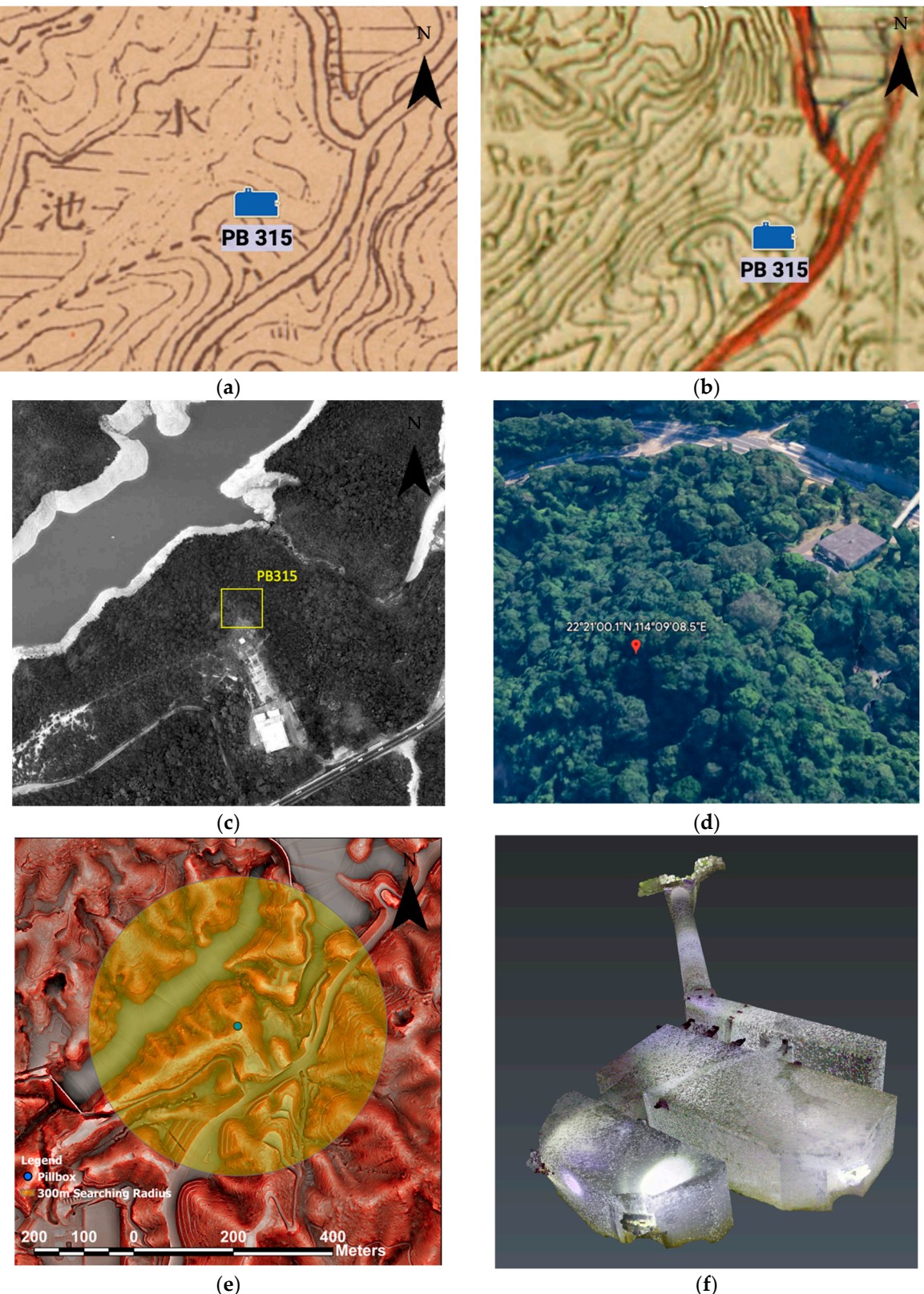

**Figure 9.** Image analysis of PB315. (**a**) 1939 Japanese military maps. (**b**) 1941 Japanese military map. (**c**) PB315 aerial photo in 1964 at 2700 ft (No. 5153). (**d**) Google Earth of PB315 in 2017. (**e**) 300 m searching radius from PB315 on RRIM. (**f**) The 3D terrestrial laser scan.

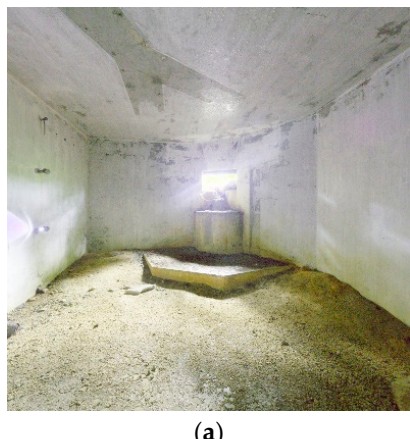 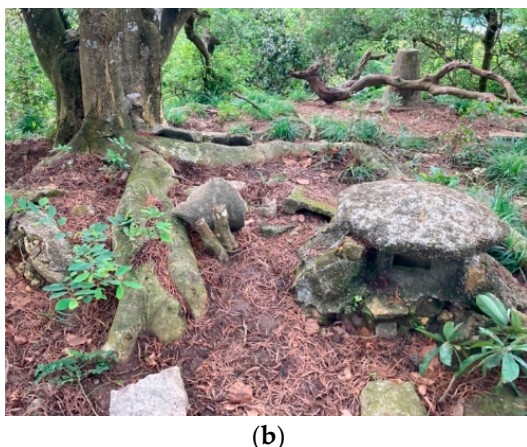

(**a**) (**b**)

**Figure 10.** Site photos of PB315. (**a**) Interior of PB315. (**b**) Mushroom-shaped air vent of PB315.

### 4.4. Modelling of Airborne DTM-RRIM and Ground-Based TLS Scanning

The integration of airborne DTM-RRIM and ground-based TLS models provides a unique opportunity for archaeologists and the public to revisit the battlefields of the 1940s, enabling a deeper analysis of military structures. Since both DTM-RRIM and terrestrial 3D point cloud models were already geo-referenced, using GIS software, coloured point clouds of military structures were overlaid on DTM-RRIM raster images. This integration model can be generated as a real 3D augmented reality and virtual reality of the battlefield in 1941. This can make the audience feel as if they were physically on the battlefield of WWII to 'feel' this often-forgotten part of HK history. Figure 15 shows the point cloud deliverables of four types of WWII features on the RRIM (3D). For battlefield archaeologists, the integration model can be employed for further spatial analysis of military structures. On a macroscopic level, the relationship between the Defence Scheme of the British Army, the network of Japanese tunnels during WWII, and the terrain of Hong Kong can be examined. On a microscopic level, the purpose and major flaws of different military lines of defence can be analysed, including the viewshed of military structures, shooting angles, and firing coverage. One of the primary functions of point cloud deliverable is 3D measurement. These deliverables provide a digital twin of real-world military structures, enabling researchers to extract any profile and 3D measurement from the product. For the general public's convenience in browsing, the 3D textured lightweight mesh model with rendering can then be generated. Textured meshes that overlay real images captured alongside them with an LTS point cloud are produced to generate photorealistic models. This format has a smaller file size than the point cloud model but retains a high degree of detail, making it more suitable for on-screen and website visualization for educational and public engagement purposes.

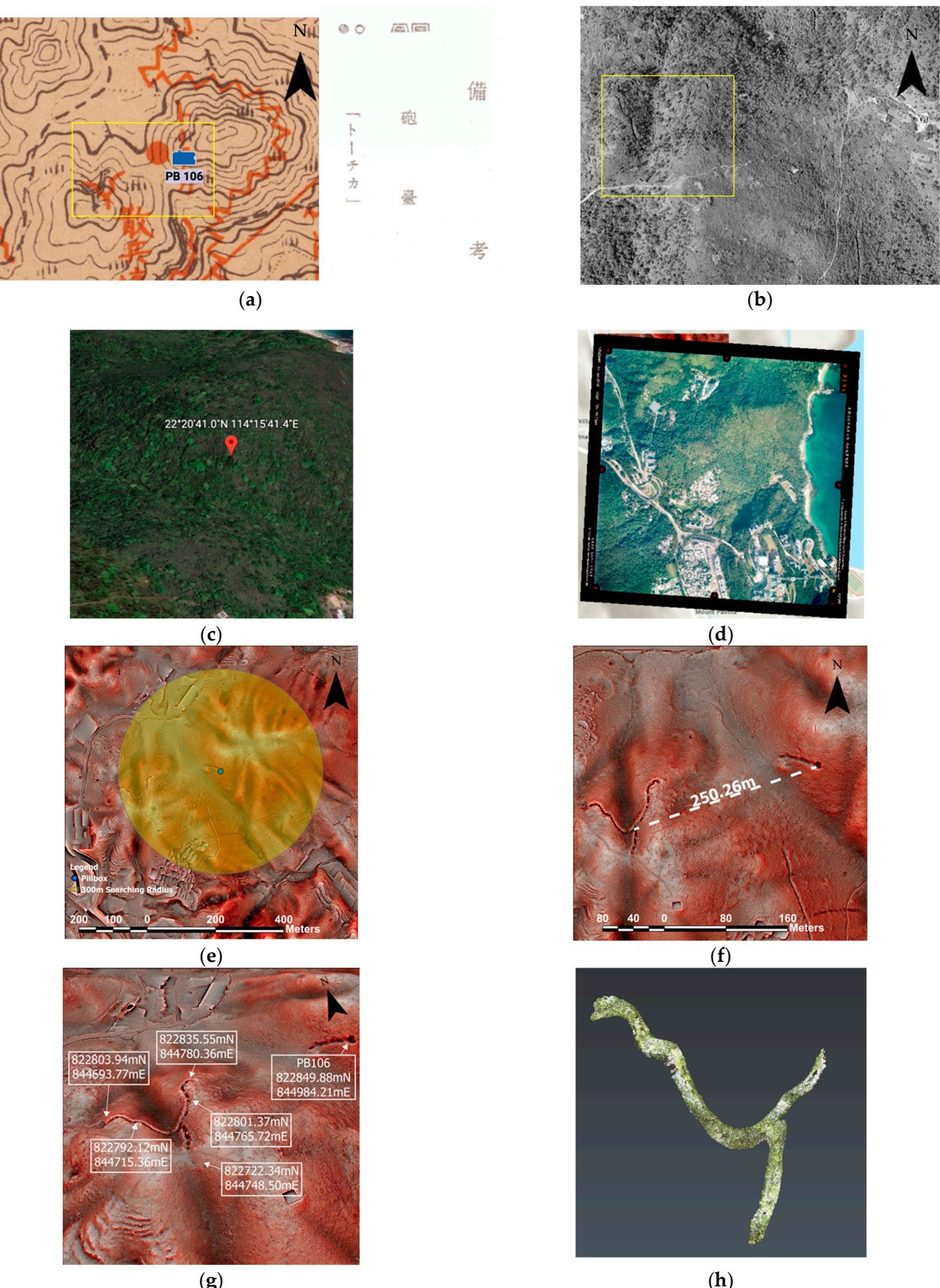

**Figure 11.** Image analysis of PB106. (**a**) 1939 Japanese map near PB106 with legend. (**b**) PB106 aerial photo in 1963 at 3900 ft (No. 9741). (**c**) Google Earth of PB106 in 2021. (**d**) PB106 aerial photo in 2016 at 6000 ft (no. CS62946). (**e**) 300m Searching Radius from PB106 on RRIM. (**f**) RRIM with 2020 Li-DAR. (**g**) Coordinates of Feature near PB106. (**h**) 3D terrestrial laser scanning model of the 'Y'-shaped Trench near PB106.

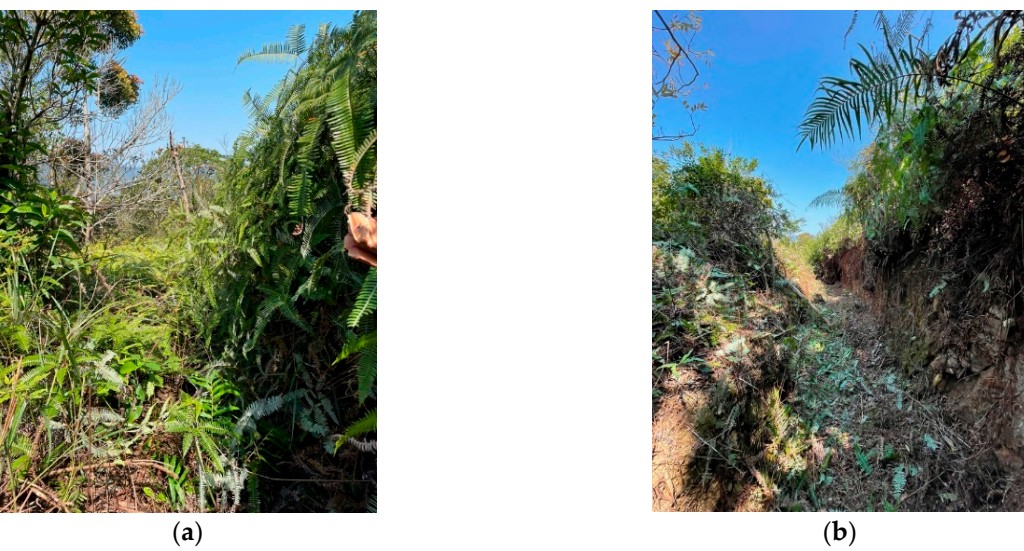

**Figure 12.** Site photos of the Y-shaped trench near PB 106. (**a**) Y-shaped trench near PB 106 before vegetation clearing. (**b**) Y-shaped trench near PB 106 after vegetation clearing.

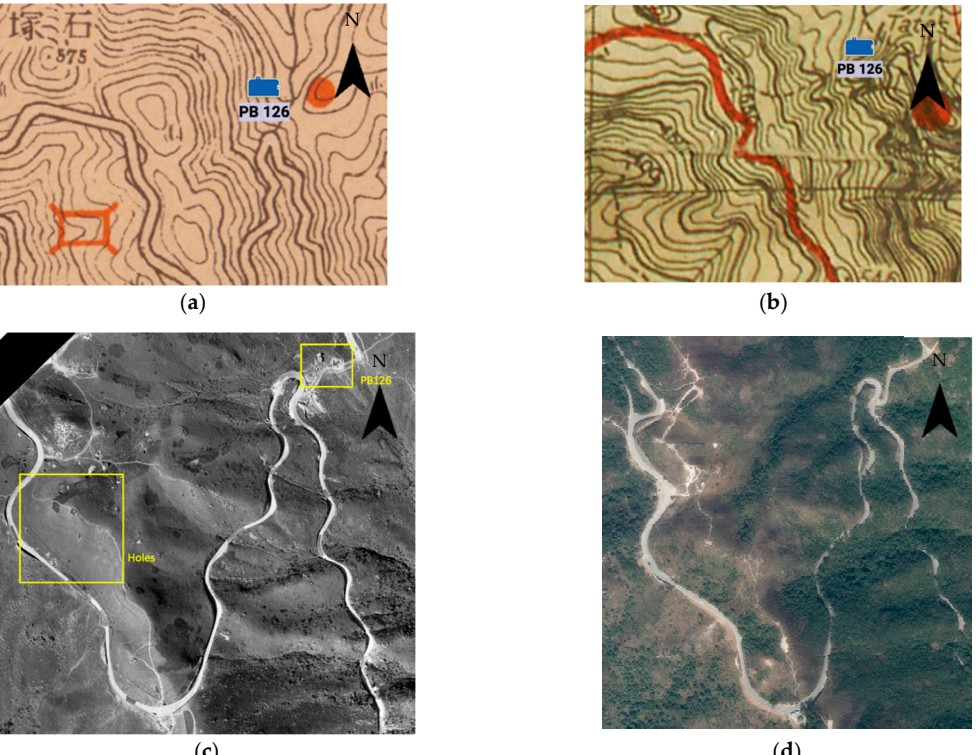

**Figure 13.** *Cont.*

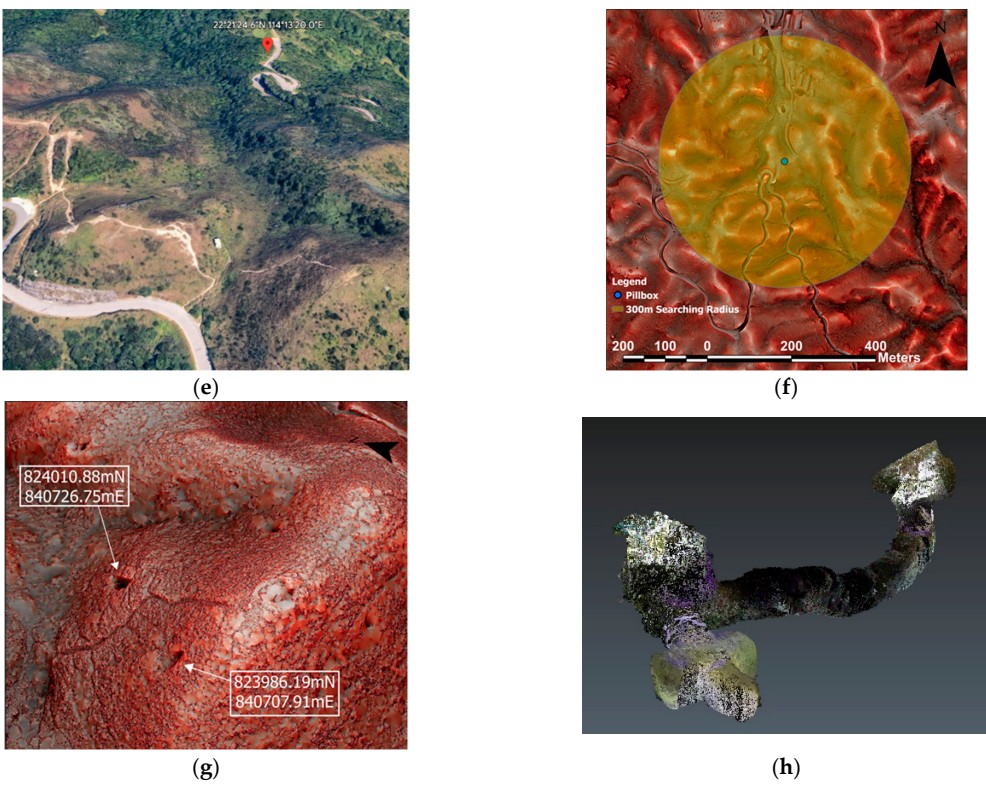

(e)

(f)

(g)

(h)

**Figure 13.** Image analysis of PB126. (**a**) 1939 Japanese military maps. (**b**) 1941 Japanese military map. (**c**) PB126 aerial photo in 1963 at 3900 ft (no. 9648). (**d**) PB126 aerial photo in 2021 at 6900 ft (no. E139265C). (**e**) Google Earth of PB126 in 2017. (**f**) The 300 m searching radius from PB126 on RRIM. (**g**) War holes of PB126 on RRIM (3D). (**h**) The 3D terrestrial laser scanning model of tunnel and cave near PB126.

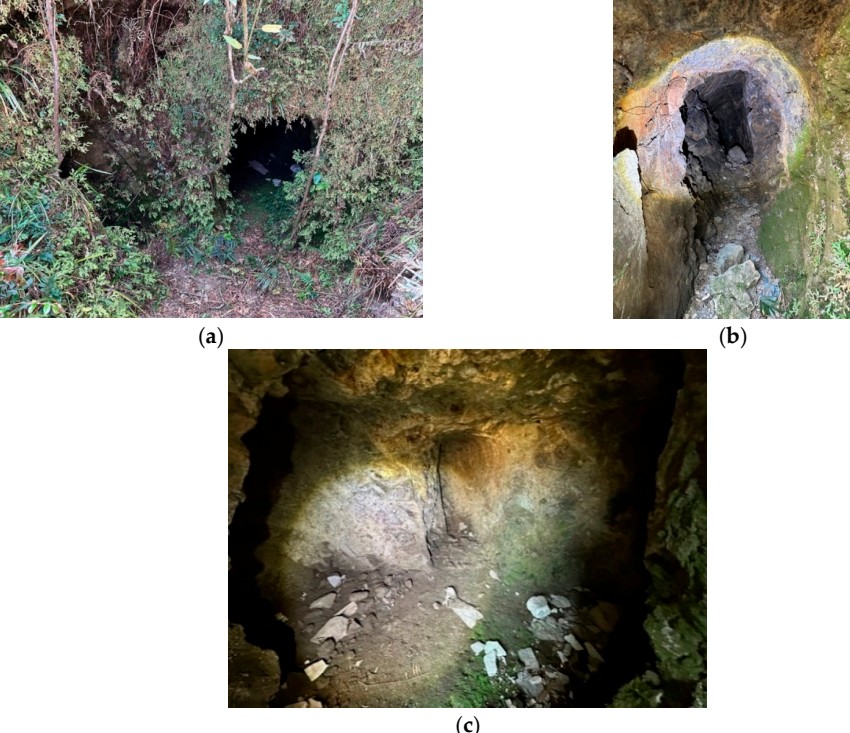

(a)

(b)

(c)

**Figure 14.** Site photos of the tunnel and cave near PB126. (**a**) An overview of the tunnel and cave near PB126. (**b**) Tunnel near PB126. (**c**) Cave near PB126.

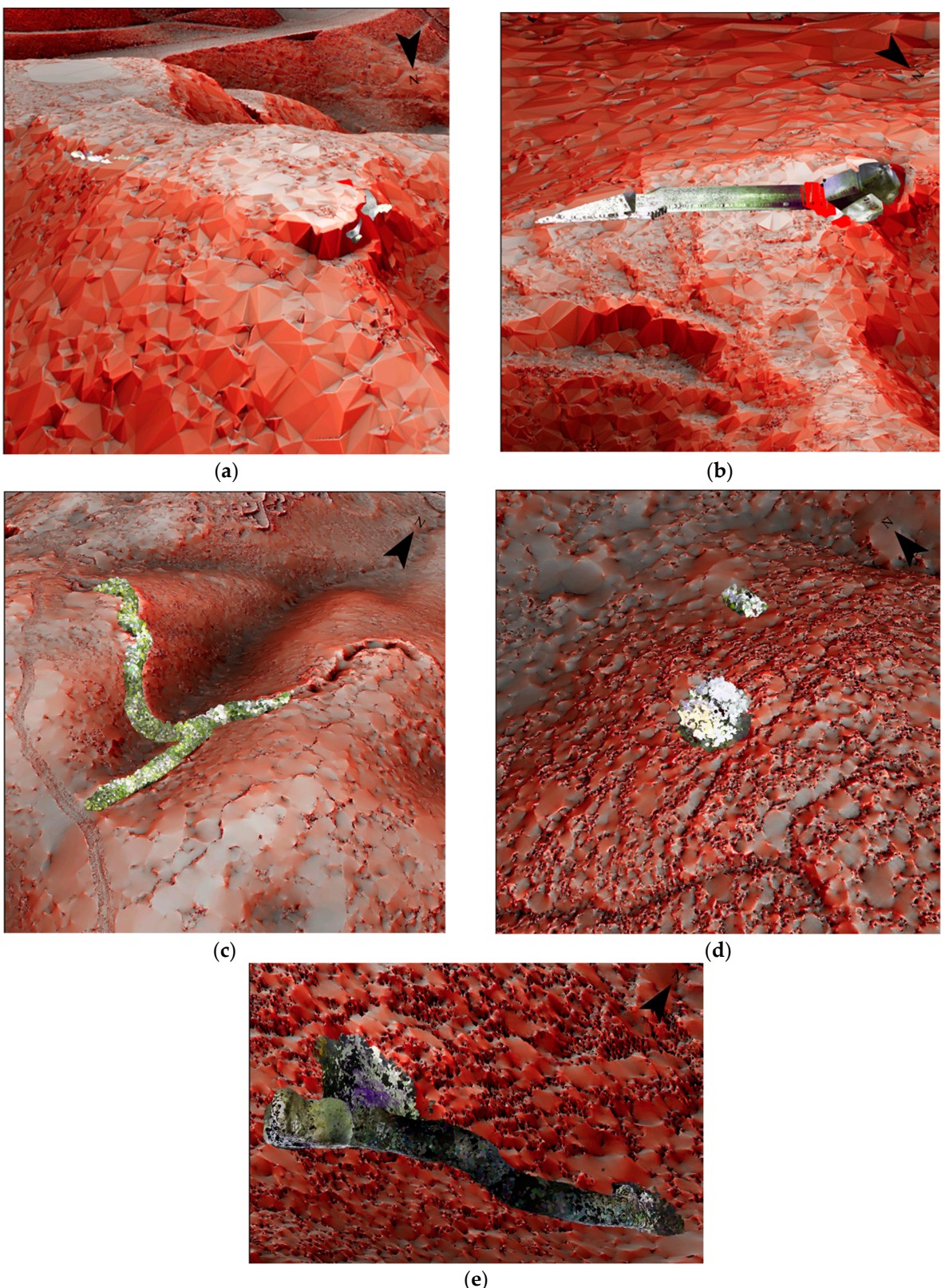

**Figure 15.** The point cloud deliverables of four features on RRIM. (**a**) PB315 above the surface on RRIM (3D). (**b**) PB315 below the surface on RRIM (3D). (**c**) The Y-shaped trench near PB106 on RRIM (3D). (**d**) Tunnel and cave near PB126 above the surface on RRIM (3D). (**e**) Tunnel and cave near PB126 below the surface on RRIM (3D).

## 5. Conclusions

This paper aims at presenting a more systematic operation by using LiDAR, laser, and GNSS-RTK technologies to improve the archaeological examination approach. The introduced technology-empowered methodology is highly reliable and accurate because point cloud data for visualisation are provided objectively without biases. In the traditional investigation method, archaeologists rely on photo and map analyses. As old maps are 2D and hand drawn, map distance, features, and bearing are always inaccurate. Archaeologists have to interpret maps, which may create biases. Unlike conventional methods, DTM-RRIM visualises objective information because point cloud data are collected using LiDAR for mapping the rugged terrain that is always overgrown with vegetation. Historical document analysis is fundamental to archaeological research as it defines the project's background, information, and prospective outcomes. Desktop study allows defining of the region of interest since accurate locations can be obtained for DTM-RRIM generation. Analysing terrain features using DTM-RRIM facilitates efficiency and effectiveness of deploying exploration teams for historical and archaeological studies. It raises the hit rate for archaeological findings as unnatural topographic features can be emphasised through algorithms. Combining this with aerial photos and old maps, which are the references for the archaeological features confirming their existence, points of interest can be defined, or a region of interest can be more precise and accurate.

In addition, different survey technologies are adopted in the field after defining the region of interest. GNSS-RTK is indispensable for storing the coordinates of the targeted features, and it can be the cross-reference for the DTM-RRIM's coordinates leading the way of exploration Geo-referenced laser-scanned 3D models visualise the entire remains for the public. The 3D mesh models can be formed after scanning, which can be demonstrated to the citizens for raising their awareness about history and historical conservation. In this project, three unknown or known features were successfully identified, which were the Y-shaped trench near PB106 and the tunnel and cave near PB126. These features are not widely reported in many documentary records. The field survey was meant to verify the existence of all three features that were identified on DTM-RRIM, and it gave no error when the expedition team visited with GNSS-RTK. This proved the effectiveness and success of the DTM-RRIM and the GNSS-RTK as the backbones of this technology-driven investigation method of WWII history.

Concerning the increase of social awareness in heritage protection in Hong Kong, this research enables interested parties, students, or NGOs to systematically understand the undiscovered features or structures hidden in the forest/vegetation to historical interpretation in three tiers as mentioned in the abstract. First, historical documentaries had to be studied for understanding the site backgrounds and potential discoveries. Then, aerial photos, old paper maps, and Google Earth were applied to understand the history/human geography. Finally, geo-spatial technologies such as airborne/terrestrial LiDAR and GNSS-RTK (hardware), the mesh and RRIM models (data processing), and GIS (platform) were blended to enrich and enhance the first two stages of studies and provide a new dimension of understanding of WWII in HK. Since the point cloud collected by the CEDD is open to the public, interested parties can adopt this scientific investigation method and study these historical relics, preserve heritages, and bring influence in heritage impact assessment. This approach can greatly draw people's attention through its collaborative nature of different disciplines. STEAM workshops and exhibitions can be arranged to demonstrate the process of how heritage remains can be discovered for the sake of restoring the lost parts of WWII history in Hong Kong. The acronym STEAM, encompassing science, technology, engineering, arts, and mathematics, can be properly elucidated as an innovative educational approach that empowers students to engage in immersive inquiry, constructive dialogue, and critical thinking throughout their journey of one or more particular aspect(s) in STEAM. It is hoped that the next generations of Hong Kong and other parts of the world will be inspired in the future to adopt the geo-spatial technologies and the work-

flow as outlined in this paper in understanding the lost and long-forgotten war-related and archaeological remains.

**Author Contributions:** Conceptualization, K.F.-C.S, C.-H.P., W.W.L.L. and D.K.-W.C.; methodology, K.F.-C.S. and C.-H.P.; software, K.F.-C.S. and C.-H.P.; validation, W.W.L.L. and D.K.-W.C.; formal analysis, K.F.-C.S. and C.-H.P.; investigation, K.F.-C.S., C.-H.P., W.W.L.L., D.K.-W.C. and C.-M.K.; resources, C.-M.K.; data curation, K.F.-C.S., C.-H.P., W.W.L.L., D.K.-W.C. and C.-M.K.; writing—original draft preparation, K.F.-C.S.; writing—review and editing, K.F.-C.S., C.-H.P. and W.W.L.L.; visualization, W.W.L.L.; supervision, W.W.L.L.; project administration, W.W.L.L.; funding acquisition, W.W.L.L. All authors have read and agreed to the published version of the manuscript.

**Funding:** This research 'Unfolding the lost WWII heritage: promotion of geo-spatial and geophysical technologies' was funded by the Innovation Technology Fund–General Support Scheme (GSP), matching fund from Research Institute of Land and Space of The Hong Kong Polytechnic University and the in-kind industrial sponsor Camptopia Limited (previously OurVillas Co. Ltd.) (ITF/GSP/043/22).

**Data Availability Statement:** Restrictions apply to the availability of these data. Data was obtained from Civil Engineering Development Department (CEDD) and Lands Department (LandsD) from HKSAR Government, and are available from CEDD at https://sdportal.cedd.gov.hk/#/en/ (accessed on 6 June 2023) and LandsD at https://www.geodetic.gov.hk/en/satref/satref.htm (accessed on 6 June 2023), respectively with the permission of CEDD and LandsD.

**Acknowledgments:** We would like to pay tribute to those who perished and were killed in action for their countries in WWII in the Battle of Hong Kong. In particular, we are most grateful to the open airborne LiDAR data from the Geotechnical Engineering Office (GEO) of Civil Engineering and Development Department (CEDD) of the HKSARG and the open-data and GNSS-RTK service from the Lands Department of the HKSARG. Lastly, we are thankful to Anthony So for the inspiration for the use of airbone LiDAR data on heritage investigation.

**Conflicts of Interest:** The authors declare no conflict of interest.

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
