# Peer review of "Unfolding WWII Heritages with Airborne and Ground-Based Laser Scanning"

_heritage, doi:10.3390/heritage6090325_

Round 1

Reviewer 1 Report

the paper presents an approach to building a digital archive presenting to the public and experts WWII heritage in the Hong Kong area. The approach is implemented with a historical GIS platform.

High-resolution 3D data have been acquired with several campaigns; those data have been studied in detail, using different cutting-edge visualization filters, to detect the remains of military infrastructures and to map all those data in an HGIS.  

The paper applies different visualization filters for the detection of trenches and fortifications on digital maps and discusses the pro and cons of the different methods.

Multiple field surveys (terrestrial laser scanning) have been subsequently done, on the base of the knowledge produced by the Tier 2 project phase. These have increased the quality and resolution of the data related to the selected structures; the data produced have been integrated with the previous GIS data.

The paper is a good example of an application of modern technologies to the archeological study of a mid-size geographic context.

No major innovation in terms of the technologies used, but a good example of a practice & experience paper.

Minor comments:

-        Page 2: Line was consisted  line consisted

-        figure 13 is not very easy to read, even zooming on the digital copy of the paper; would it be possible to improve the quality of this rendering, making it more easy to understand/comprehend?

none

Author Response

Response is attached

Reviewer 2 Report

The paper explores the possibilities of using multivariate remote sensing tools in combination with historical cartography and terrestrial methods to document Second World War relics in the Hong Kong countryside. Unfortunately, in its present form, the article comes across as a study written in a less than scholarly manner, lacking innovation in its methodological approach. At the same time, the article is structured outside the basic statutes of the MDPI (citation apparatus, chapter titles, acknowledgement of the journal Heritage in the page footers is dedicated to the 2021???). There are a number of inaccuracies and missing references to appropriate literature. The topic of the paper is certainly an interesting and useful proposal, but the study needs to be reworked into a coherent description of the methodology and, above all, to conclude the study with a broader synthesis, which this paper lacks (see other sub-comments). Unfortunately, I cannot recommend the paper for publication in this form and suggest that it be substantially revised.

Figure 1 lacks source

Chapter 2.1 is written in a very superficial way with respect to the topic of prospection of the archaeological and historical heritage, the introductory paragraph is redundant. The second paragraph presents random projects, that have no relevance to the topic. It completely ignores the synthesising monographs and scientific studies published in recent years in the field of archaeology and the usage of lidar data as the prospection method (R. Opitz, D. Cowley, S. Campana, M. Doneus, etc.). There is no indication of the density at which the data were acquired, or what form of interpolation was used. According to the image documentation, the lidar data is very coarse - I recommend trying a different form of interpolation and including also the Local Relief Model visualisation for more effective feature identification.

The title of chapter 2.1 is mentioned twice in the paper.

-        Chapter 2.1. unacceptable description of lidar visualisation, authors cite unoriginal papers, no citations to original sources for application of lidar visualisation to historical and archaeological data (R. Hesse, B. Štular, BJ. Devereux, R. Opitz, R. Bennett etc.). I suggest that this chapter be substantially revised and expanded.

-        Chapter 3.2. Why was a grid size of 5 cm chosen? How was this resolution chosen? Is it the maximum resolution?

-        Chapter 3.1 –„Analysis of Aerial Photographs Aerial Photographs“ – the colon is missing

-        Figure 6 is very cluttered. I suggest putting the descriptions of features 1-4 directly into the figures as a legend.

-        Placing figures 11 and 12 directly behind each other is confusing.

-        The quality of the map output is at print screen level (at first glance it appears that the output are really just print screens without any form of appropriate attributes for professional use). There is a need to create higher resolution exports of map views directly in appropriate software (especially in the case of GIS outputs). There is no scale or north orientation anywhere - unacceptable outputs.

-        Occasional missing points, align entries in Table 1.

-        Chapter 4 should be renamed to Results.

-        In general, the article does not follow the MDPI citation format - references should be numbered in the text.

           What are STEAM workshops? – Sentence in the chapter Conclusions.: 'STEAM workshops and exhibitions can be organised to demonstrate the process of discovering heritage remains for the purpose of restoring the lost part of Hong Kong's Second World War history'

The article completely lacks a general synthesis of the issue. Beyond the partial case studies, it is not clear how much of the presented area has been methodologically analysed and how many newly discovered features related to WWII events have been captured. The use of lidar data and visualisation algorithms to enhance feature detection in archaeology is now a very common practice. I see the overlap of this work into a methodological innovation in the possibilities of applying ground scanning and documenting the real condition of individual features and then visualising them in 3D models, combined with the underlying DEM relief from lidar data. However, this process is described very superficially here, and the output from the ground scanner is visualised in very little detail - it would certainly be interesting to display the 3D model (or a section of it) in a larger format to show its quality. For the case studies, I suggest uploading the models to something like SketchFab and including a link in the text to view them in 3D.

Author Response

Response is attached.

Reviewer 3 Report

The research methodology is well described and designed. The approach to the research topic is interesting. The bibliography is not very updated. The surveying techniques used are already well consolidated in the literature. The bibliographical references should be updated.

Author Response

Response is attached.

Reviewer 4 Report

Dear authors,

the paper is well structured and organized.

the state-of-the-art and the methodology are clearly explained and the results demonstrate de useful of LiDAR and TLS for the historical analyse of the landscape: it is really interesting to understand how is it possible to study and reconstruct through these techniques a war scenery.

Only few missprints:

p.3. "2.1. Visualisation Methods and GIS in Tier 2 (Air-Borne LiDAR)": change the ordering number with "2.2. Visualisation Methods and GIS in Tier 2 (Air-Borne LiDAR)"

p. 4: "2.1.1. Positive Topographic Openness and Negative Topographic Openness": change the ordering number with "2.2.1. Positive Topographic Openness and Negative Topographic Openness"

p. 5: "2.1.1. Sky-View Factor". change the ordering number with "2.2.2. Sky-View Factor"

p. 5: "2.1.1. Red Relief Image Map" change the ordering number with "2.2.3. Red Relief Image Map"

If is it possible, put only image that is divided in two pages in only one page  (i.e. figs. 5, 6, 9, 10, 11, 12)

Author Response

Response is attached.

Round 2

Reviewer 2 Report

In this form, the paper can be approved and published. I thank the authors for their comments and corrections.